# Study of SarA by DNA Affinity Capture Assay (DACA) Employing Three Promoters of Key Virulence and Resistance Genes in Methicillin-Resistant *Staphylococcus aureus*

**DOI:** 10.3390/antibiotics11121714

**Published:** 2022-11-28

**Authors:** Byungchan Kim, Hong-Ju Lee, Sung-Hyun Jo, Min-Gyu Kim, Yeonhee Lee, Wonsik Lee, Wooseong Kim, Hwang-Soo Joo, Yun-Gon Kim, Jae-Seok Kim, Yung-Hun Yang

**Affiliations:** 1Department of Biological Engineering, College of Engineering, Konkuk University, Seoul 05029, Republic of Korea; 2Department of Chemical Engineering, Soongsil University, 511 Sangdo-dong, Seoul 06978, Republic of Korea; 3Department of Biotechnology, College of Engineering, Duksung Women’s University, Seoul 01370, Republic of Korea; 4Department of Pharmacy, School of Pharmacy, Sungkyunkwan University, Suwon-si 16419, Republic of Korea; 5College of Pharmacy, Graduate School of Pharmaceutical Sciences, Ewha Womans University, Seoul 03760, Republic of Korea; 6Department of Laboratory Medicine, Kangdong Sacred Heart Hospital, Hallym University College of Medicine, Seoul 05355, Republic of Korea; 7Institute for Ubiquitous Information Technology and Applications, Konkuk University, Seoul 05029, Republic of Korea

**Keywords:** MRSA, DACA, *sarA*, virulence factors, clinical strain

## Abstract

Methicillin-resistant *Staphylococcus aureus* (MRSA), one of the most well-known human pathogens, houses many virulence factors and regulatory proteins that confer resistance to diverse antibiotics. Although they have been investigated intensively, the correlations among virulence factors, regulatory proteins and antibiotic resistance are still elusive. We aimed to identify the most significant global MRSA regulator by concurrently analyzing protein-binding and several promoters under same conditions and at the same time point. DNA affinity capture assay (DACA) was performed with the promoters of *mecA*, *sarA*, and *sarR*, all of which significantly impact survival of MRSA. Here, we show that SarA protein binds to all three promoters. Consistent with the previous reports, *ΔsarA* mutant exhibited weakened antibiotic resistance to oxacillin and reduced biofilm formation. Additionally, production and activity of many virulence factors such as phenol-soluble modulins (PSM), α-hemolysin, motility, staphyloxanthin, and other related proteins were decreased. Comparing the sequence of SarA with that of clinical strains of various lineages showed that all sequences were highly conserved, in contrast to that observed for AgrA, another major regulator of virulence and resistance in MRSA. We have demonstrated that SarA regulates antibiotic resistance and the expression of various virulence factors. Our results warrant that SarA could be a leading target for developing therapeutic agents against MRSA infections.

## 1. Introduction

*Staphylococcus aureus* is one of the most common bacteria in humans; however, it may cause serious infections such as pneumonia, osteomyelitis or other tissue infections in immunocompromised individuals [1,2]. Methicillin-resistant *Staphylococcus aureus* (MRSA) was first reported in 1961 and named thus [3]. Since then, MRSA resistant to various antibiotics has emerged, complicating the treatment of MRSA infections, and becoming a global healthcare issue [4]. MRSA expresses many different virulence factors to adapt to new environments and infect other hosts [5]. Factors related to virulence and resistance include penicillin-binding protein 2a (PBP2a), biofilm, staphyloxanthin, and motility [6,7]. These resistance- and virulence-related genes are regulated by regulatory proteins that either directly or indirectly bind to their promoter regions to control gene expression [8,9].

As a representative regulator in *S. aureus*, the accessory global regulator (*agr*) is well known as a quorum-sensing regulatory protein and has been intensively studied [10]. *agr* is located adjacent to a two-component signal transduction system composed of RNAII and RNAIII [11]. AgrA activates these two transcriptional units and forms a positive feedback loop [12]. In particular, RNAIII and AgrA regulate various virulence factors, such as phenol-soluble modulins (PSM), hemolysins, enterotoxins, *mecA*, and Panton-Valentine leucocidin (PVL) as well as various toxins and proteases [12].

Another gene related to virulence, staphylococcal accessory regulator A (*sarA*), encodes the SarA protein, a 14.7 kDa DNA binding protein that regulates the expression of various virulence factors [13]. SarA not only regulates hemolysins, fibronectin-binding proteins, proteases, and biofilm formation, but it also directly interacts with *agr* regulon [14,15,16]. SarA regulates the expression of *agr* by binding between the P2 and P3 promoters of the *agr* regulon [17]. The sar family of genes directly or indirectly regulates various target genes through protein expression [18]. The *sarR* gene encodes SarR, a 13.6 kDa DNA-binding protein [19]. SarR binds to P1, P2, and P3 of the upstream promoter regions of *sarA*, and negatively regulates the expression of the *sarA* gene [20]. Additionally, *sarR*, like *sarA*, also regulates the transcription of *agr* [21].

Identifying the regulators for specific targets is essential, in particular, for understanding complex regulatory systems. To achieve this, we previously developed a DNA affinity capture assay system in *Streptomyces* producing antibiotics with very complex regulatory systems and identified several novel regulatory proteins regulating antibiotic production and morphological changes [22,23]. Like *Streptomyces* sp., MRSA seems to be a good candidate for DACA as it has a complex regulatory system controlling antibiotic resistance and virulence.

Since the study of regulatory systems requires considerable effort and time to confirm the function of each regulator, we tried to identify the significant global regulators in MRSA by binding to several promoters and analyzing them concurrently under the same conditions and at the same time point. DACA enables the identification of the principal regulators binding together on various promoters.

In this study, we attempted to identify regulators that bind to the promoters of *mecA*, *sarA*, and *sarR* genes (Figure 1A). *mecA* encoding PBP2a, which plays a significant role in antibiotic resistance [24], was selected to identify the principal regulators of resistance and *sarA* and *sarR* were used to search for regulators involved in virulence. In addition, we attempted to identify novel functions of *sarA* by comparing the phenotype of MRSA in the presence and absence of *sarA*. By comparing the gene sequences with those of clinical MRSA isolates, we also provided insight into the main role of SarA.

## 2. Results

### 2.1. DNA Affinity Capture Assay

The sequences for *mecA*, *sarA*, and *sarR* promoters were from the genome of *S. aureus* USA300 JE2 using PCR and 3′-biotinylated primers (Appendix A). Following the DNA binding experiments with the promoters and cell lysates (explained in Section 4), the samples were prepared for further analysis. Each protein sample was analyzed using Nano-liquid chromatography mass spectrometry (nLC-MS/MS) through tryptic digestion, and protein identification was performed by comparing the sequent HT scores (Figure 1A and Appendix A). As there were many non-specific binding proteins which were known to have DNA binding affinity, known specific regulatory proteins were selected in each promoter (Figure 1B and Appendix A). Although many novel proteins bound to each promoter, an excessive amount of research and time would have been necessary to elucidate their functions; therefore, we only targeted the previously well-known regulator SarA for this study.

Atl, SarA, CodY and SarR proteins bound to the *mecA* promoter. Atl, SarA, CodY, SarR, PSM and SigB proteins bound to the sarA promoter. Atl, SarA, SarR, MgrA and Spx proteins bound to the sarR promoter (Appendix A). Among several proteins, SarA, SarR, and Atl bound to all three promoters. Atl is a bifunctional protein with amidase and endo-β-N-acetylglucosaminidase domains and is involved in cell separation and stress-induced autolysis and contributes to bacterial pathogenesis [25]. The glucosaminidase domain exhibits DNA-binding activity [26]. SarR, a SarA paralog, represses SarA expression. Inactivation of *sarR* affects several target and regulatory genes [21]. This result shows that previous individual studies worked well even in simultaneous studies. Among them, SarA, one of the Sar family proteins, is known to be significantly involved in the regulation of several virulence factors in MRSA (Table 1). In addition, SarA directly or indirectly regulates over 100 genes [27]. Therefore, we attempted to verify the novel functions of SarA and demonstrate the role of global regulators using the DACA method based on their binding affinity with promoters [28]. We obtained a transposon mutant in which the *sarA* gene encoding the SarA protein was deleted and compared it with the JE2 strain from the Nebraska Transposon Mutant Library collection (https://app1.unmc.edu/fgx/tools, accessed on 20 February 2020).

### 2.2. Effect of sarA on Antibiotic Resistance and Virulence Factors

First, we compared the antibiotic resistance of *ΔsarA* mutant and JE2 strains. JE2 showed a minimum inhibitory concentration (MIC) of 32 μg/mL against oxacillin, whereas *ΔsarA* showed an MIC of 4 μg/mL. This confirmed that resistance to oxacillin was significantly reduced, although it is still methicillin resistance according to the MIC criteria (Figure 2A). Generally, MRSA forms biofilm to protect itself from the effects of antibiotics [40]. However, at 0.5 μg/mL oxacillin concentration, biofilm formation by *ΔsarA* was slightly decreased compared with that of JE2 (Figure 2B). Additionally, biofilm production was almost completely inhibited at 64 μg/mL in JE2, whereas the inhibition occurred at 2 μg/mL in *ΔsarA*.

Hla, also known as alpha-toxin, is one of the toxins secreted by *S. aureus* bound to the eukaryotic cell membrane [41]. In particular, hla binds to red blood cells and forms pores to induce hemolysis [42]. *ΔsarA* significantly suppressed the expression of hla (Figure 2C).

In addition to antibiotic resistance, biofilm formation and α-hemolysin production, which have been reported previously, PSM production was first measured by MALDI-TOF in *ΔsarA* [15,16]. PSM is an amphiphilic peptide with properties similar to those of surfactants and affects the spread of MRSA [43]. PSM binds to formyl peptide receptor 2(FPR 2) on the surface of various immune cells and promotes an inflammatory response. At higher concentrations, alpha-type PSMs act as toxins that lyse host cells [44,45]. JE2 and *ΔsarA* samples were prepared (as explained in Section 4), and PSM quantification was performed for comparison. Almost all types of PSM were completely inhibited in *ΔsarA* (Figure 2D). Based on previous reports, a possible scenario for the correlation between PSM production and Agr could be attributed to the regulation of *agr* by SarA. Inactivation of *sarA* could decrease the expression of *agr*, resulting in a decrease in PSM levels. Although this cascade was expected, the direct evidence of decreased PSMs has not previously been shown.

### 2.3. Alteration in Motility and Staphyloxanthin Production of ΔsarA

*S. aureus* while non-motile, can move by secreting the surfactant-like substance PSM [46,47]. The motility of *S. aureus* significantly influences biofilm formation and host colonization [46]. Both factors have a great impact on its infectivity and on cell survival. Therefore, we compared the motility of JE2 and *ΔsarA* and found that the motility of *ΔsarA* was significantly reduced compared with that of JE2 (Figure 3A). Considering that there was no significant difference in the cell growth of JE2 and *ΔsarA* in the previous growth results, it can be seen that the motility was significantly reduced (Figure 2A). The previous results also showed that *sarA* had a significant effect on PSM production, supporting the reason for the significant decrease in motility. In addition, CsrA, which binds to the flhDC transcript, stabilizing the molecule and ultimately contributing to bacterial motility and RNA-binding properties exhibited by SarA was hypothesized to control bacterial motility of bacteria [48,49]. Although it was not confirmed at molecular level, our result suggested the direct evidence of relationship between SarA and motility.

Staphyloxanthin is a golden carotenoid pigment produced by *S. aureus* and has a role as a virulence factor. This pigment has antioxidant properties against host’s reactive oxygen species (ROS) and also has resistance to neutrophils, thus protecting cells from the host’s immune system [50]. We compared the staphyloxanthin production of JE2 and *ΔsarA* at 6 h intervals. In the case of *ΔsarA*, the production of staphyloxanthin decreased compared with that of JE2 at each time point (Figure 3B). In addition, as shown in Figure 3A, the color of the *ΔsarA* colony changed slightly to white compared with that of JE2. These data present the properties of global regulators pertaining to various functions such as PSM production, motility, and staphyloxanthin production, which were not previously well-known for SarA.

### 2.4. Effect of sarA on Proteomic Changes of MRSA

To analyze the effect of *sarA* gene-on the virulence of MRSA at the intracellular protein level, comparative proteomic analysis was performed using JE2 and *ΔsarA* mutants. Principal component analysis (PCA) showed changes in proteins that were clearly distinguished according to the expression level of the *sarA* gene (Figure 4A). A total of 1079 proteins were identified by untargeted proteomic analysis, of which 93 proteins were significantly upregulated (log_2_ (fold-change) >0.58, adjusted *p*-value <0.05), and 117 proteins were significantly down-regulated (log_2_ (fold-change) < −0.58, adjusted *p*-value <0.05) (Figure 4B). Firstly, the proteomics data showed that the expression of the Agr system (AgrC, AgrB) and PSM (psmα4) was significantly down-regulated, whereas that of the repressor of toxin (Rot) was up-regulated in the *ΔsarA* mutant (Figure 4C). The reduced expression levels of AgrC and AgrB are consistent with previous reports that SarA directly and positively regulates the *agr* gene [51]. Furthermore, the agr system increases RNAIII transcription, which inhibits the translation of Rot and promotes the transcription of PSMs [52]. Therefore, the decreased level of PSMα4 and increased level of Rot could be attributed to the down-regulated agr system. Secondly, the expression of extracellular proteases, such as Aur, SspB, and Staphopain A (ScpA) was significantly up-regulated in the *ΔsarA* mutant (Figure 4C). Several studies have reported that these proteins negatively affect biofilm formation [53,54]. SarA is a repressor of extracellular proteases, including Aur, SspB, and ScpA, and these extracellular proteolytic enzymes accumulate in the culture supernatant of the *ΔsarA* mutant [53,54,55]. Therefore, it is suggested that overexpression of Aur, SspB, and ScpA may have contributed to reduced biofilm formation in the *ΔsarA* mutant (Figure 2B and Appendix A). Lastly, the production of PBP2a was specifically down-regulated in the *ΔsarA* mutant (Figure 4C). PBP2a encoded by *mecA* has a low affinity for β-lactam antibiotics, including methicillin, and confers the antibiotic resistance to *S. aureus* known as MRSA [56]. The eight-fold decrease in the MIC of oxacillin in the *ΔsarA* mutant compared with that in the JE2 wild-type may be due to a decrease in the expression of PBP2a (Figure 2A). In addition, the MIC value for the *ΔsarA* mutant decreased four and eight-fold, respectively, against the other beta-lactam antibiotics ampicillin and penicillin (Appendix A). Therefore, SarA appears to play an important role in antibiotic resistance related PBP2a expression. These results suggest that SarA may positively regulate *mecA* expression directly or indirectly and may be the reason for the decreased expression of PBP2a in the *ΔsarA* mutant.

### 2.5. Analysis of SarA Sequence in Various Clinical Strains

Numerous mutations have been identified in gene sequences of various MRSA lineages [57]. From these mutations, virulence of MRSA is either enhanced or weakened [58]. However, there may be genes that rarely mutate in various lineages of MRSA, suggesting that these genes have a significant role and influence on the cell [59]. To evaluate this, we compared the amino acid sequences of SarA proteins using whole-genome sequencing, targeting 10 clinical strains of various lineages isolated from patients (Table 2 and Figure 5). In the case of SarA, no change in sequence was observed in the strains from any of the lineages (Figure 5). When we expanded the search for SarA amino acid sequences in the NCBI database, most sequences were highly conserved, and only one sequence (at site 53 from phenylalanine to leucine (Sequence ID: WP_031921994.1)) or two sequence changes (at site 34 from phenylalanine to leucine and site 53 from phenylalanine to leucine (Sequence ID: WP_031921994.1)) were reported. Although the effect of these mutations on SarA is unclear, the number of mutations seems very small in proportion to the amino acid size of SarA, which is 113 amino acid. This reaffirms the importance of SarA. Several mutations were detected in AgrA in the clinical samples and among the AgrA sequences in NCBI. This suggests that SarA is a more general and global regulator in *S. aureus* existing at the upper level of AgrA.

Among the strains from the clinical samples that were analyzed for PSM production, HL16278, HL17064, HL18807, and HL21008 showed no production of PSMαs, PSMβs and δ-toxins but showed AgrA mutation (Table 2); this could be attributed to fact that PSM production is directly regulated by the Agr system [12]. However, HL18883 had AgrA mutation, yet still produced PSMs. The exact reason for this odd response is not known. This might suggest the existence of another control mechanism apart from Agr itself. Thus, studying the upper level of AgrA and other SarA-like upper regulators is of prime importance.

## 3. Discussion

In this study, we identified various proteins that affect the expression of each promoter using DACA. Among the various proteins identified, SarA, a key protein of the sar family, binds to all three promoters [28]. In addition, the function of SarA in *S. aureus* is diverse, including the regulation of key virulence factors. Though individual reports provided ample insight into the role of SarA, the overall perspective provided by the DACA approach enabled simultaneous evaluation of various regulators, and we gained new information regarding regulatory proteins controlled by key promoters.

To confirm the importance of *sarA* in influencing many virulence factors of MRSA, AgrA and SarA sequences pertaining to several lineages of MRSA isolated from patients were compared. The analysis showed various mutations in *agr*, which was previously known to affect various virulence factors, but none in *sarA*, despite the comparison being made among various MRSA lineages. Proteomics analysis showed that the expression of *mecA* was decreased in the *sarA*-deficient mutant. Currently no known study is available to answer if *sarA* regulates *mecA*; however, the *mecA* expression was reduced in the *sarA* -deficient mutant. Hence, further research is needed to determine if any relationship exists between the two genes. In addition, the extracellular proteases Aur, SspB, and ScpA were overexpressed in the *sarA*-deficient mutant. Decreased expression of these genes may be another reason for the weakened antibiotic resistance of *ΔsarA* because both genes are involved in biofilm formation and the expression of PBP2a [24,60].

## 4. Materials and Methods

### 4.1. Strains, Media and Culture Conditions

*S. aureus* USA300 JE2 and transposon mutant strain NE1193 (*ΔsarA*) were obtained from the Nebraska Transposon Mutant Library collection (https://app1.unmc.edu/fgx/tools, accessed on 20 February 2020). Tryptic soybean broth (TSB) was used to culture cells. A single colony of the strain from a TSB agar plate was inoculated with 5 mL of TSB medium for pre-culture. One percent (*v*/*v*) of the cell culture suspension was inoculated into TSB for subsequent cell cultivation at 37 °C in a shaking incubator (200 rpm) [61].

### 4.2. Preparation of Protein Extracts

JE2 cells were grown in 50 mL of TSB with shaking (200 rpm) at 37 °C for 24 h and harvested via centrifugation (3000× *g* for 20 min). The cell pellet was washed twice using phosphate buffer saline (PBS, pH 7.4) and suspended in Buffer II (20 mM HEPES, pH 7.8, 10% *w*/*v* glycerol, 100 mM KCl, 0.05 mM EDTA, 0.5 mM dithiothreitol (DTT), 0.01% Nonidet P-40, 1 mM PMSF, 1 mM benzamidine, 1 g/mL leupeptin, and 1 g/mL pepstatin with 0.5 M NaCl). We tried to maximize the protein yield by optimizing the sonication time and confirmed that the protein concentration was maximized under the following conditions. The obtained cells were disrupted using an Ultrasonic Processor (Sonics & Materials, Newtown, CT, USA, amp 30%, 10 s pulse at 15 s intervals) for approximately 1 h. Thereafter, the cell extract was centrifuged to separate the supernatant containing the proteins. The concentration of the protein extract was determined using Bradford assay (Bio-Rad, Hercules, CA, USA), and the samples were stored at −80 °C [22].

### 4.3. DNA-Affinity Capture Assay

The schematic procedure of the DACA is outlined in Figure 1A. The upstream regions of *mecA*, *sarA* and *sarR* including the promoter sequence, were amplified using polymerase chain reaction (PCR) and biotinylated primers (Appendix A). The PCR products were purified using gel purification and amplified using PCR. Concentrations of the purified DNA were determined using a SpectraMax M2 microplate reader (Marshall Scientific, Hampton, NH, USA). We found the optimal PCR product concentration to prevent non-specific binding and set the concentration to about 300 ng/μL. Streptavidin MagneSphere Paramagnetic Particles (Promega, Madison, WI, USA) were washed thrice in Buffer I (20 mM Tris pH 8.0, 20 mM MgCl_2_ and 200 mM KCl). Biotinylated PCR products were washed with the beads (100 pmol/mg of beads) and incubated for 30 min at room temperature in Buffer I, followed by the addition of biotin (100 μg/mL) and another 15 min of incubation. One milliliter of a pre-mixed solution of cell extract (500 μg/mL) and sheared salmon sperm DNA (ssDNA, 0.1 mg/mL) were added. The mixture was incubated for 15 min on ice to prevent non-specific binding. Next,100 μL of the bead solution (0.5 mg beads/100 μL Buffer II) was added and incubated for 1 h at room temperature. Beads were washed four times with Buffer II and once with 2M NaCl, followed by incubation for 10 min in a 90 °C heat block for the separation of DNA and protein. Then the supernatant was collected [22].

### 4.4. Protein Precipitation Assay

Trichloroacetic acid (TCA) solution (10:7 (*w:w*); TCA: distilled water) was mixed with the protein sample in a 1:4 (*v:v*) ratio. After incubation at 4 °C for 10 min, centrifugation (3000× *g* for min) was performed to obtain a white precipitated protein pellet. The protein pellet was washed twice with cold acetone. Proteins were obtained by evaporating the acetone in a 95 °C heat block for 10 min [62].

### 4.5. Liquid Chromatography-Tandem Mass Spectrometry (LC-MS/MS) Analysis

Protein pretreatment was performed for LC-MS/MS analysis. The protein samples were incubated in 10 mM DTT for 30 min at 56 °C. The samples were cooled at room temperature with additional iodacetamide (final concentration 20 mM) for carboxyamidomethylation. For in-solution digestion, sequencing grade trypsin (Sigma Aldrich, St. Louis, MO, USA) in 100 mM NH_4_HCO_3_ was added, and the samples were digested overnight at 37 °C. Nano High Resolution LC-MS/MS spectrometry (Thermo Fisher Scientific, Waltham, MA, USA) was performed using an electron and nano electron ionization spray with a quadruple and FT orbitrap mass analyzer. Peptide ions were detected in the full scan mode with a maximum at 2000 *m*/*z*. The maximum mass resolution was 140,000 at *m*/*z* 200. Proteins were identified by searching MS/MS spectra data against the Proteome Discoverer protein database.

### 4.6. Antimicrobial Susceptibility and Biofilm Formation

Antimicrobial susceptibility and biofilm formation were assessed in a 96-well plate. The dispensing and serial dilution of 200 µL of TSB medium containing oxacillin in each well were performed automatically using a liquid handler (Integra, Le Locle, Switzerland). The cells were precultured and inoculated at a concentration of 1% (*v*/*v*). 96-well plates were incubated at 37 °C for 24 h without shaking. Optical density was measured at 595 nm using a 96-well plate reader (Thermo Fisher Scientific) [61]. To analyze biofilm formation, the supernatant of the culture broth was carefully removed. Biofilm fixation was performed using methanol and air-dried overnight. The biofilm of each well was stained with 200 µL crystal violet 0.2% (in methanol, (*w*/*v*)) for 5 min. The staining dye was carefully removed and washed twice with distilled water. Finally, absorbance was measured at 595 nm using a 96-well microplate reader [63].

### 4.7. Analysis of PSM and α-Hemolysin

PSMs in the culture supernatants were quantified using liquid chromatography-mass spectrometry (LC-MS) after centrifugation at 3000× *g* for 20 min, as described previously [64,65]. The supernatant (5 µL) was injected into a C8 column (ZORBAX SB-C8, 2.1 mm × 30 mm, 3.5 µm, Agilent, Santa Clara, CA, USA) linked to a ZQ 2000 LC-MS system (Waters, Milford, MA, USA), and PSMs were separated based on hydrophobicity. Trifluoroacetic acid (TFA) 0.05% (*w*/*w*) in double distilled water and TFA 0.05% (*w*/*w*) in acetonitrile were used for gradient elution. PSMs were quantified by integrating the corresponding peaks in the extracted ion chromatograms based on the mass-to-charge ratios of the two highest ion peaks with different charge states for each PSM, as described previously [65].

α-hemolysin (Hla) was quantified using western blot immunoassay as previously described, with some modification [66]. After SDS-PAGE of the culture supernatants, the separated proteins were blotted onto NC membranes using iBlot 2 Dry Blotting System (Thermo Fisher Scientific). The blotted membranes were blocked with Intercept^®^ (TBS) Blocking Buffer (LI-COR, Lincoln, NE, USA) for 1 h. Anti-staphylococcal α-toxin antibody rabbit serum (Sigma, USA) was diluted in blocking buffer at a 1:2000 ration and incubated overnight at 4 °C. The membranes were washed five times with 0.1% Tween 20-mixed Tris buffered saline (TBST) and incubated in the dark for 1 h in a blocking buffer containing IRDye 800 CW goat anti-rabbit antibody (LI-COR, USA) at a 1:10,000 dilution ratio. After five washes with TBST, the membranes were scanned using an Odyssey CLx Imager (LI-COR, USA), and Hla expression levels were quantified using Image Studio software (LI-COR, USA).

### 4.8. Motility Assay in a Soft Agar Plate

To determine the change in motility caused by deletion of *sarA*, we conducted a previously reported soft agar assay [46,47,67]. Pre-cultured cells were centrifuged and resuspended in an equal volume of PBS, and 2 μL of the mixture were spotted onto the center of a 0.24% TSB agar plate. The plates were then incubated for 10 h at 37 °C. All experiments were performed in triplicate.

### 4.9. Staphyloxanthin Extraction and Quantification

Cells were grown in 5 mL of TSB with shaking (200 rpm) for 6, 12, 18, and 24 h at 37 °C and harvested by centrifugation (3000× *g* for 20 min). The cell pellet was washed once with PBS and centrifuged. After the supernatant was completely removed, the pellet was resuspended in 500 μL methanol and incubated at 55 °C for 20 min for staphyloxanthin extraction. Following centrifugation, 200 μL of staphyloxanthin in methanol was obtained. Extracts containing staphyloxanthin were filtered through a 0.2 μm syringe filter (Chromdisc, Hwaseong, Korea) to prevent contamination with the cell pellet. The amount of pigment in each sample was determined immediately by measuring the optical density at 470 nm using a plate reader (Thermo Fisher Scientific) [61,68].

### 4.10. Preparation Samples for Proteomics Analysis

To prepare samples for intracellular proteomic analysis, we modified and used the filter-aided sample preparation (FASP) method [69,70]. *S. aureus* JE2 and the *ΔsarA* mutant cells at the stationary phase were centrifuged and the cell pellets were washed twice with PBS at 4 °C and resuspended in RIPA buffer (Thermo Fisher Scientific) containing a 0.1% protease inhibitor cocktail (Sigma-Aldrich, USA). The cells were lysed by ultra-sonication (35% amplitude, 10 s pulse, 10 s rest, 15 min) on ice using a probe sonicator (Sonics & Materials, Inc., Newtown, CT, USA). Supernatants of the cell lysates were acquired after centrifugation at 3000× *g* for 10 min at 4 °C. Protein concentration was measured using a bicinchoninic acid protein assay kit (BCA assay; Thermo Fisher Scientific) according to the manufacturer’s instructions. For protein reduction, 1 M DTT was added to a final concentration of 50 mM and incubated for 5 min at 95 °C. Reduced proteins (100 μg) were transferred to 30 k filter units (Microcon; Millipore, Burlingame, MA, USA). UA buffer (8 M urea in 0.1 M Tris-HCl, pH 8.5) was added to the device to obtain a final volume (proteins +UA buffer) of 500 μL and centrifuged at 10,000× *g* for 30 min at 20 °C. The protein samples were washed thrice with 200 μL of UA buffer. The protein samples were alkylated with 100 μL of 0.05 M iodoacetamide (IAM) in UA buffer and incubating in the dark at room temperature for 20 min. After the samples were centrifuged at 10,000× *g* for 15 min at 20 °C, the proteins were washed twice with 100 μL of UA buffer and then twice with a digestion buffer (100 μL 0.05 M Tris-HCl, pH 8.5). Then, 40 μL of trypsin (0.05 μg/μL) was added to the samples in the digestion buffer and digested for 20 h at 37 °C. The peptides were collected by centrifuging 260 μL of digestion buffer. The peptides were desalted on Pierce Peptide Desalting Spin Columns (Thermo Fisher Scientific) according to the manufacturer’s instructions. The purified samples were dried with a speed vac. (Vision Scientific, Seoul, Korea) and stored at −80 °C until needed for further analysis.

### 4.11. Bottom-up Proteomic Analysis by LC-MS/MS

Proteomic analysis was performed as previously described [71]. Briefly, the dried samples were dissolved in 2% ACN + 0.1% *v*/*v* formic acid in water to obtain a final peptide concentration of 1 mg/mL. The peptide samples were separated using nano-HPLC, Ultimate 3000 RSLCnano LC system (Thermo Fisher Scientific) and analyzed using Q Exactive Hybrid Quadrupole-Orbitrap (Thermo Fisher Scientific) equipped with a nano-electrospray ionization source. First, 2 μg of peptides were trapped in an Acclaim PepMap 100 trap column (100 μm × 2 cm, nanoViper C18, 5 μm, 100 Å, Thermo Fisher Scientific). Subsequently, 98% of solvent A (0.1% *v*/*v* formic acid in water) was used to wash the column at a flow rate of 4 μL/min for 6 min. After washing, the samples were separated at a flow rate of 350 nL/min. An Acclaim PepMap 100 capillary column (75 μm × 15 cm, nanoViper C18, 3 μm, 100 Å, Thermo Fisher Scientific) was used for separation. The LC gradient was as follows: 0 min, 2% B; 30 min, 35% B; 40 min, 90% B; 45 min, 90% B; 60 min, 5% B, where B stands for Solvent B (0.1% formic acid in acetonitrile). Solvents A and B were used for this step. The ion spray voltage was 2100 eV. MS data were collected using the Xcaliber software. The Orbitrap analyzer scanned the precursor ions with a mass range of 350–1800 *m*/*z* and resolution of 70,000 at *m*/*z* 200. For collision-induced dissociation (CID), up to the 15 most abundant precursor ions were selected (Top15 method). The normalized collision energy was 32. Protein identification and label-free quantification (LFQ) were performed using the MaxQuant software [72,73]. Proteins were identified by searching the MS and MS/MS data of peptides against the *S. aureus* USA300 in the proteome database (September 2021) in Uniprot. Quantified protein data were processed and statistically analyzed using Perseus software, and significance was determined using the adjusted *p*-value [74].

## 5. Conclusions

Using DACA, we revealed the global interaction between SarA and various promoters, confirming the important roles SarA plays in virulence. We confirmed the conservation of SarA sequence based on clinical *S. aureus* isolates and attempted to deduce the direct impact of *sarA* inactivation. These results demonstrate the effectiveness of DACA in elucidating the mechanism underlying promoters of resistance and virulence related genes in *S. aureus*. We believe that DACA can be implemented in further studies on different promoters and targets to accelerate the research on resistance. It is particularly useful to study regulatory proteins in strains that have complex regulatory networks. Although this study focused only on SarA, analyzing the interactions among three promoters and various proteins has unveiled a list of previously unknown regulators which could be harnessed for further research

## Figures and Tables

**Figure 1 antibiotics-11-01714-f001:**
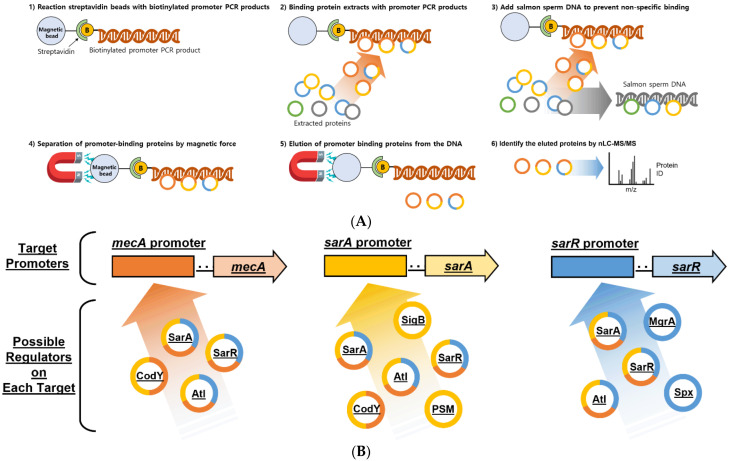
Schematic figure of DNA affinity capture assay (DACA) and identification through Nano-liquid chromatography mass spectrometry (nLC-MS/MS) of various proteins that bind to target pro-moters (**A**). In DACA, each promoter region of *mecA*, *sarA*, and *sarR* genes was amplified using PCR before use. (**B**) Identification of possible regulators on each target promoter through nLC-MS/MS.

**Figure 2 antibiotics-11-01714-f002:**
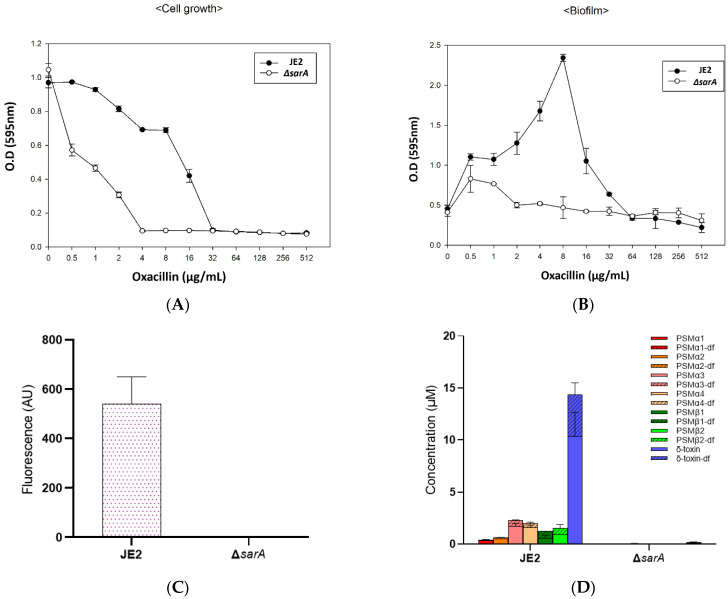
Antibiotic resistance of *ΔsarA* and changes in the expression of various virulence factors. The cell growth and biofilm formation of the JE2 and *ΔsarA* strains were measured in varying oxacillin concentrations. Optical densities of cells were compared at 595 nm (**A**). After removing the supernatant of the culture medium, the degrees of biofilm formation were compared at O.D of 595 nm through methanol fixation and crystal violet staining (**B**). JE2 and *ΔsarA* were presented in black and white circles and statistical analysis was performed by applying ANOVA with the level of significance set at 5% (**A**,**B**). Hla of JE2 and *ΔsarA* strains were quantified and compared by western blot immunoassay. Hla signal was hardly detected in *ΔsarA* unlike in JE2 (**C**). Each type of PSM was quantified and compared by LC-MS analysis. All types of PSM production were significantly reduced in *ΔsarA* (**D**).

**Figure 3 antibiotics-11-01714-f003:**
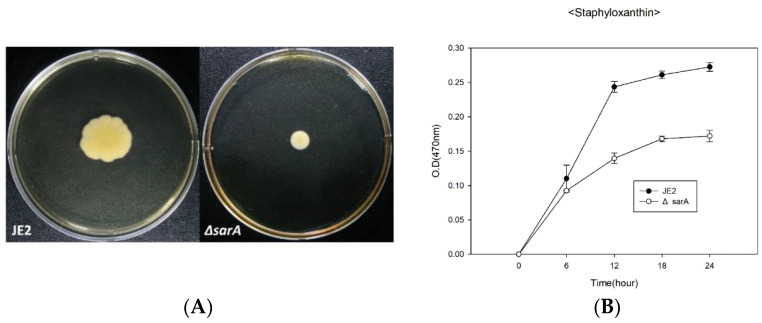
Motility and staphyloxanthin alteration of *ΔsarA*. Motility tests were performed by culturing for 10 h on soft agar TSB plates. The tests were performed in triplicate with similar results (**A**). Cell pellets of samples harvested at each time were extracted through methanol, and the extracted staphyloxnathin were compared by measuring absorbance at 470 nm (**B**). JE2 and *ΔsarA* were presented in black and white circles and statistical analysis was performed by applying 240 ANOVA with the level of significance at 5% (**B**).

**Figure 4 antibiotics-11-01714-f004:**
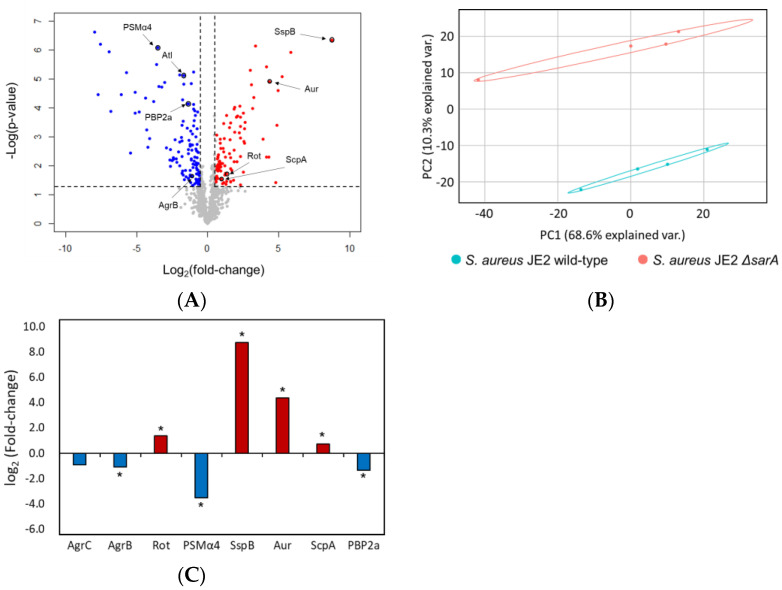
Comparison of proteomic changes between *S. aureus* JE2 and *ΔsarA* mutant. (**A**) Volcano plot of proteomic changes for JE2 and *ΔsarA* mutant. Red color indicates relatively high expression in *ΔsarA* mutant and blue color indicates relatively low expression in *ΔsarA* mutant. Proteins with fold change > 1.5 and with adjusted *p*-value < 0.05 were considered statistically significant (*n* = 4). (**B**) Principal component analysis (PCA) plot of proteomic analysis for JE2 and *ΔsarA* mutant. (**C**) Changes in the expression of proteins related to the virulence factor of *S. aureus*. The red bar indicates relatively high expression, and the blue bar indicates relatively low expression in the *ΔsarA* mutant compared with that of the wild type. The asterisk marks (*) indicate adjusted p-value < 0.05.

**Figure 5 antibiotics-11-01714-f005:**
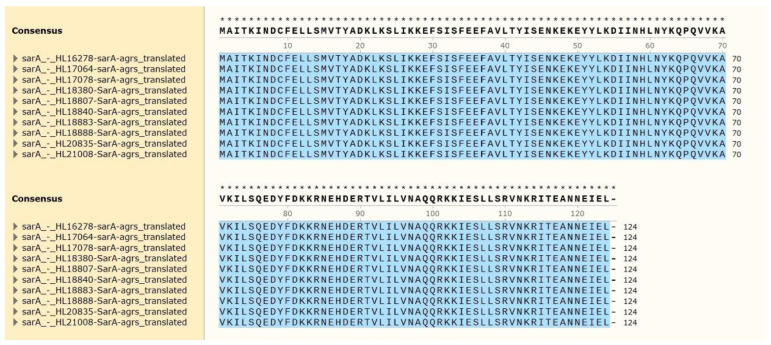
Alignment of SarA proteins in clinical samples.

**Table 1 antibiotics-11-01714-t001:** Genes regulated by SarA and their effects.

Gene	Regulation	Function	Ref.
*cap*	+	Improves staphylococcal toxicity by interfering with phagocytosis.	[29]
*clfB*	+	Promotes adhesion to immobilized epidermal cytokeratin.	[30]
*cna*	−	Bacterium-host adherence and in immune evasion.	[31]
*fnbA, B*	+	Attachment to fibronectin	[15]
*spa*	−	Ability to bind to the Fc region of IgG	[15]
*sdrC*	+	Promotes adhesion of bacteria to surfaces and biofilm formation.	[27]
*femA*	+	Cytoplasmic protein required for the expression of methicillin resistance and involved in the biosynthesis of the staphylococcal cell wall	[27]
*atl*	−	Peptidoglycan hydrolase that breaks down peptidoglycan to release daughter cells	[27]
*icaADBC*	+	Polysaccharide intracellular adhesion and poly-N-acetylglucosamine polysaccharide production and biofilm formation	[32]
*hla*	+	Tissue invasion and form pores in host cell membrane	[15]
*hlb*	+	Selectively cytotoxic to monocytes	[33]
*hld*	+	Wide spectrum of cytolytic activity	[27]
*hlgABC*	+	Causes toxic shock syndrome with toxin shock syndrome toxin 1	[27]
*seb*	+	Superantigens with severe toxic effects on the immune system	[34]
*geh*	+	Glycerol ester hydrolase in *S. aureus*	[33]
*aur*	−	Effectively inhibits phagocytosis and bacterial death by neutrophils.	[35]
*tst*	+	Toxins act as superantigens, activating very large numbers of T cells and generating an overwhelming immune-mediated cytokine storm	[36]
*lukF, lukS*	+	Causes leukocyte lysis and tissue necrosis	[37]
*sspA*	−	Degrades the Fc region of immunoglobulins and leads to partial loss of antigenic determinants of the antibody.	[35]
*sspB*	−	Causes imbalance in the homeostasis of host immune cells in the inflamed tissue	[35]
*scpA*	−	Breaks down elastin and has broad specificity	[38]
*agr*	+	Global regulator, which includes secreted virulence factors and surface proteins	[27]
*mecA*	+	Involved in beta-lactam antibiotics activity	[39]

**Table 2 antibiotics-11-01714-t002:** Sequence analysis of AgrA and SarA, with PSM production for each clinical isolate.

Clinical Isolates	SCCmec	Spa Type	MLST	CC	Agr Type	AgrA Mutation	AgrA	SarA	NCBI Reg.No	PSMα1	PSMα2	PSMα3	PSMα4	PSMβ1	PSMβ2	δ-toxin
HL16278	IVA	T324	ST72	CC8	I	S202N	238 aa	124 aa	CP080564-CP080565	−	−	−	−	−	−	−
HL17064	II	T2460	ST5	CC5	II	I238K	238 aa	124 aa	CP080560-CP080561	−	−	−	−	−	−	−
HL17078	IV	T008	ST5863	CC8	I		238 aa	124 aa	CP080556-CP080559	+	+	+	+	+	−	+
HL18380	IV	T008	ST8	CC8	I		238 aa	124 aa	CP080553-CP080555	+	+	+	+	+	−	+
HL18807	II	T2460	ST5	CC5	II	I238NNNKIIKSVNGVFNCKSCWILTR	260 aa	124 aa	CP080552	−	−	−	−	−	−	−
HL18840	IVA	T324	ST72	CC8	I		238 aa	124 aa	CP080551	+	+	+	+	+	+	+
HL18883	II	T9353	ST5	CC5	II	D137N, I238NNNKIIKSVNGVFNCKSCWILTR	260 aa	124 aa	CP080550	+	+	+	+	−	−	+
HL18888	II	T002	ST5	CC5	II		238 aa	124 aa	CP080548-CP080549	+	+	+	+	−	−	+
HL20835	II	T002	ST5	CC5	II		238 aa	124 aa	CP080566-CP080567	+	+	+	+	−	−	+
HL21008	II	T9353	ST5	CC5	II	D137N, I238NNNKIIKSVNGVFNCKSCWILTR	260 aa	124 aa	CP080562-CP080563	−	−	−	−	−	−	−

## Data Availability

Data is contained within this article and Appendix A.

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
