# Peer review of "Study of SarA by DNA Affinity Capture Assay (DACA) Employing Three Promoters of Key Virulence and Resistance Genes in Methicillin-Resistant Staphylococcus aureus"

_antibiotics, 2022, doi:10.3390/antibiotics11121714_

Round 1
Reviewer 1 Report
Brief Summary
Kim et al. embark on a journey to identify a significant global MRSA regulator by analyzing the proteins bound to promotors in the MRSA genome. Specifically, they focused on SarA protein binding to the mecA, sarA, andsarR promotors, which they followed up with characterizing ΔsarA mutant.
Significance
Emerging antibiotic resistance limits the usably of existing antibiotics. Of interest is methicillin-resistant Staphylococcus aureus (MRSA) a nosocomial pathogen which causes a difficult to treat staph infection. Better understanding of MRSA is urgently needed, especially more insight into the regulation of antibiotic resistance and of expression of virulence factors. This type of data can help control of common pathogens, by helping to minimze the emergence of resistance to antibiotics. Furthermore, such data can help provide new therapeutics for multi drug resistant pathogens in the future.
Recommendations:
I recommend accepting this paper with minor revisions for publication at the Antibiotics Journal. I am listing below minor suggestions for clarifying details described in this review. I am not recommending any additional experiments.
Notes on the text:
General:
- As there were many other proteins detected to bind the promotors you studied, I think your paper will benefit from framing this study as the study of SarA, rather than “Rediscovery of SarA by DNA affinity capture assay”.
- Please include more details in the figure legends, including description of the assays, such as: cell growth, levels of secreted PSM evaluated using MALDI-TOF, staphyloxanthin levels evaluated by … Especially for figures 2 and 3.
- Please refer to your supplementary data in the main text. Such as the full list of promotor binders are in table S…
Line 98: “Although many novel proteins bound to each promoter, plenty of research and time would have been necessary to elucidate their functions; therefore, we only selected the previously well-known regulators for this study. “ As there are many other proteins binding to these promotors, I think your paper can benefit from centering SarA rather than stating that SarA is a strong hit in this screen.
Line 161: Figure 2 legend. Please rewrite the legend for figure 2. The dexcription for C and D panels was switched. Please add the detailed description for each of the panels. For example, “The cell growth (A) and biofilm formation (B) of the JE2 and ΔsarA strains were quantified in varying oxacillin concentrations. JE2 and ΔsarA and presented in black and white circles. “ “ PSM production measured by MALDI-TOF… “
Table 2 is illegible due to small font size.
Line 249: “we compared the amino acid sequences of SarA proteins using whole- 249 genome sequencing, targeting 10 clinical strains of various lineages isolated from patients 250 (Table 2).” I think you meant to refer to figure 5 rather than table 2.
Line 267: where is the data for PSM production in clinical strains. I think you meant to refer to table 2.
Reviewer 2 Report
The authors described MS-based promoter sequence binding regulating protein identifications. After identification of the binding pairs, the authors turned to more experiments to directly show the effect of this regulation over growth of the strain.
While the conclusions were supported by the experiments, no new scientific discoveries were made in this study, nor did the authors demonstrate the advantage of their experimental procedure over previous settings. Does the technique bring new discovery or improve existing techniques? It would be interesting to see any updates on these perspectives.
Reviewer 3 Report
The manuscript "Rediscovery of SarA by DNA affinity capture assay (DACA) employing three promoters of key virulence and resistance genes in methicillin-resistant Staphylococcus aureus" is of great interest to a wide range of readers. The authors adequately describe the research methods, give a good introduction to the problem, and formulate the research objectives. From small remarks, I would like to note the presence in the list of references of a large number of manuscripts older than 10 years. I recommend the authors to replace some of the cited literature with more modern ones.
Reviewer 4 Report
Using a DNA affinity capture experiment, Byungchan Kim et al. have rediscovered that the Sar A protein binds to each of the three promoters, mecA, sarA, and sarR. (DACA). Additionally, they discovered limited oxacillin resistance and biofilm development in the sarA mutants, which was consistent with earlier observations. Overall, the research is quite interesting, and I have a few recommendations or comments for the authors to make to make the paper even better.
Comments:
· Line 36 write our results
· I did not find the figure 1 (b), include the n LC-MS/MS result in figure
· Line 193 in place of triplicated it can be written as triplicate
Round 2
Reviewer 2 Report
Thank you to the authors for the response. The authors argued that this study is to show that their DACA assay can be used for a potentially high-throughput discovery of new promoters, and discovery of new targets would take years to verify, thus would be out of the scoop.
If a quick careless search is done with `dna affinity capture assay`, some recent studies using DACA were published and suggested high-throughput assays can be done, and new DNA-binding functional proteins were identified, for example, Appl Microbiol Biotechnol. 2016 May;100(10):4495-509. doi: 10.1007/s00253-016-7306-1. This study was published in a journal with a similar level back in 2016. This is just the first paper I can see fit the requirements of 1) demonstrating high throughput protocol works; 2) report some new findings. You can keep on going with the list based on the searching results.
Based on the above mentioned observations, unfortunately I cannot agree with the authors and have to keep my previous suggestion if there is additional resubmission to current or other journals with similar level impact: 1) the author needs to show there is improvement on the protocol; 2) some new finding needs to be done.
Author Response
Response to Reviewer 2 Comments
Point 1: Thank you to the authors for the response. The authors argued that this study is to show that their DACA assay can be used for a potentially high-throughput discovery of new promoters, and discovery of new targets would take years to verify, thus would be out of the scoop.
If a quick careless search is done with `dna affinity capture assay`, some recent studies using DACA were published and suggested high-throughput assays can be done, and new DNA-binding functional proteins were identified, for example, Appl Microbiol Biotechnol. 2016 May;100(10):4495-509. doi: 10.1007/s00253-016-7306-1. This study was published in a journal with a similar level back in 2016. This is just the first paper I can see fit the requirements of 1) demonstrating high throughput protocol works; 2) report some new findings. You can keep on going with the list based on the searching results.
Based on the above mentioned observations, unfortunately I cannot agree with the authors and have to keep my previous suggestion if there is additional resubmission to current or other journals with similar level impact: 1) the author needs to show there is improvement on the protocol; 2) some new finding needs to be done.
Response 1: We appreciate reviewer’s delicate comment. We also agree with reviewer’s point in some part, especially at the level of molecular microbiology study. As SarA is well known regulator, and our teams were more focused on proteomic approach like DACA, MS based proteome data analysis and sequencing, this comment really hurt our weakness.
However, considering DACA can be improved with different conditions and expanded to different species, we think our work is also meaningful. Especially one regulator has many different functions and characteristics, targeting of same known regulator is still meaningful too.
As another reason, considering reviewer’s example paper “DNA affinity capturing identifies new regulators of the heterologously expressed novobiocin gene cluster in Streptomyces coelicolor M512” also followed our previous paper “Mass spectrometric screening of transcriptional regulators involved in antibiotic biosynthesis in Streptomyces coelicolor A3(2), (J Ind Microbiol Biotechnol, 2009 Aug;36(8):1073-83. doi: 10.1007/s10295-009-0591-2)” and commented “Unexpectedly some well-known regulatory proteins, such as the global regulators NdgR(we found it), AdpA, SlbR(we found it), and WhiA were captured in similar intensities by all four tested promoter regions” it suggested DACA technique could be used for different target and expanded to different gene clusters and further studies even in same species were also very important.
Although we did not fully show all our genetic trials on different regulators found by DACA in MRSA, we also tried to make deletion mutant of newly found regulators in MRSA, which was not easy due to clinical strains and tried to get strains from Nebraska Transposon Mutant Library library, which we found out many regulators found by DACA were missing. As a result, we had got to turn our researches on proteomic analysis and study of known regulator SarA with clinical strains which we performed genome sequencing for.
In addition, we think we have meaningful results in some points.
- As different species needs different conditions for the application of DACA, our work clearly showed the previous regulators could be rescreened by our method and DACA with our protocol could mine new regulators in MRSA.
- Application of several promoters and finding of simultaneous binding of regulators in MRSA showed the combinatorial function of regulatory proteins.
- The information on new regulators binding on the important promoters gave many different target of studies.
- Direct function of SarA on Staphlyoxhantin and motility were shown. We studied alteration in the expression of virulence factors directly or indirectly expressed by sarA We also made some new discoveries that were previously unknown. As an example, the crt operon has been known to directly regulate staphyloxanthin expression. However, in our study, sarA deletion mutants showed a specific decrease in the expression of staphyloxanthin. This indicates that sarA may be involved in the expression of various virulence factors indirectly. In addition, the reduced motility in sarA mutants can be said to be a new unknown function that sarA is involved in the infectious factor of S. aureus. Therefore, we carefully believe that these results open the possibility of further studies on the existence of various factors that indirectly affected by sarA, in addition to the previously known roles of sarA.
- We compared the effect of sarA on protein expression by proteome study and confirmed that 93 proteins were significantly up-regulated, and 117 proteins were significantly down-regulated. In this study, among these 210 proteins, we selected some proteins whose expression levels were significantly altered from known proteins. Several putative proteins not mentioned in this study were also confirmed to have altered the expression level. These results can match the putative proteins of the DACA results and will be helpful when studying the actual role of these unknown proteins. In addition, in this study, the expression level of PBP2a in the sarA mutant was decreased. In the absence of many related papers on the relationship between sarA and PBP2a, we believe that our proteomic data that sarA affects the expression of PBP2a is a meaningful result that can further strengthen the argument of the previous paper (The Global Regulon sarA Regulates β-Lactam Antibiotic Resistance in Methicillin-Resistant Staphylococcus aureus In Vitro and in Endovascular Infections, 10.1093/infdis/jiw386).
- Comparative study of SarA from clinical strains gave more emphasis on the importance of SarA even in clinical samples.
- We used more optimized protocol for aureus. Compared with previous one, we added leupeptin, another protease inhibitor, to maximize the concentration of regulatory proteins present in the cell lysate and increase the level of LC-MS/MS detection. In fact, we confirmed that the total amount of proteins yield was higher by bradford assay when leupeptin was added. The cell disruption process was also optimized. When sonication was carried out for too long, the yield was not very good because the protein itself was affected, and when the sonication time was too short, the cell of gram-positive bacteria, S. aureus, was not easily broken, so the yield was not good. Therefore, we found the optimal sonication time through several trials (Optimized time and conditions were stated at the manuscript). In addition, to prevent the possibility of non-specific binding due to excessive DNA concentration, the optimal DNA concentration was found through LC-MS/MS data identified for each DNA concentration (about 300 ng/μl). Finally, we used two methods to increase the sensitivity of LC-MS/MS in the obtained protein solution. First, we precipitated the protein obtained through trichloroacetic acid solution with modified process to increase purity of protein (DNA affinity capturing identifies new regulators of the heterologously expressed novobiocin gene cluster in Streptomyces coelicolor M512, 10.1007/s00253-016-7306-1) and prior to LC-MS/MS analysis, DTT treatment with trypsin digestion were also performed by mixing the two methods to ensure protein identification.
Although we did not reveal the function of putative regulator or hypothetical protein as reviewer mentioned, our works on SarA clearly contained new information on known SarA regulator and we think that it is the reason of other three reviewers gave us the opportunity to go further. We added some contents in the text.
We really appreciate reviewer’s time and effort on our research and expect precious comments enabled us to improve our manuscript.